# Metacognitive Strategy Training Improves Decision-Making Abilities in Amnestic Mild Cognitive Impairment

**DOI:** 10.3390/jintelligence11090182

**Published:** 2023-09-11

**Authors:** Foteini Aikaterini Pikouli, Despina Moraitou, Georgia Papantoniou, Maria Sofologi, Vasileios Papaliagkas, Georgios Kougioumtzis, Eleni Poptsi, Magdalini Tsolaki

**Affiliations:** 1Cognitive Psychology and Applications, Postgraduate Course, School of Psychology, Faculty of Philosophy, Aristotle University, 54124 Thessaloniki, Greece; 2Laboratory of Psychology, Department of Cognition, Brain and Behavior, School of Psychology, Faculty of Philosophy, Aristotle University, 54124 Thessaloniki, Greece; 3Laboratory of Neurodegenerative Diseases, Center for Interdisciplinary Research and Innovation (CIRI-AUTH), Balkan Center, Aristotle University, 10th km Thessaloniki-Thermi, 54124 Thessaloniki, Greece; gpapanto@uoi.gr (G.P.); poptsielena@gmail.com (E.P.); tsolakim1@gmail.com (M.T.); 4Day Center “Greek Association of Alzheimer’s Disease and Related Disorders (GAADRD)”, 54643 Thessaloniki, Greece; 5Laboratory of Psychology, Department of Early Childhood Education, School of Education, University of Ioannina, 45110 Ioannina, Greece; m.sofologi@uoi.gr; 6Institute of Humanities and Social Sciences, University Research Centre of Ioannina (URCI), 45110 Ioannina, Greece; 7Department of Biomedical Sciences, School of Health Sciences, International Hellenic University, 57400 Thessaloniki, Greece; vpapaliagkas@gmail.com; 8Department of Turkish Studies and Modern Asian Studies, Faculty of Economic and Political Sciences, National and Kapodistrian University of Athens, 15772 Athens, Greece; gkougioum@ppp.uoa.gr; 9Department of Psychology, School of Health Sciences, Neapolis University Pafos, 8042 Pafos, Cyprus; 101st Department of Neurology, Medical School, Aristotle University of Thessaloniki (AUTh), 54124 Thessaloniki, Greece

**Keywords:** mild cognitive impairment, decision-making, metacognitive training, analytical system, decision rules

## Abstract

Mild cognitive impairment (MCI) is associated with deficits in decision-making, which is of utmost importance for daily functioning. Despite evidence of declined decision-making abilities, research on decision-making interventions for MCI is scarce. As metacognition seems to play an important role in decision-making, the present study’s aim was to examine whether a metacognitive strategy training can improve MCI patients’ decision-making abilities. Older adults—patients of a day care center, diagnosed with amnestic MCI (n = 55) were randomly allocated in two groups, which were matched in gender, age and educational level. Τhe experimental group (n = 27, 18 women, mean age = 70.63, mean years of education = 13.44) received the metacognitive strategy training in parallel with the cognitive and physical training programs of the day care center, and the active control group (n = 28, 21 women, mean age = 70.86, mean years of education = 13.71) received only the cognitive and physical training of the center. The metacognitive strategy training included three online meeting sessions that took place once per week. The basis of the intervention was using analytical thinking, by answering four metacognitive-strategic questions, to make decisions about everyday situations. To examine the efficacy of the training, the ability to make decisions about everyday decision-making situations and the ability to apply decision rules were measured. Both groups participated in a pre-test session and a post-test session, while the experimental group also participated in a follow-up session, one month after the post-test session. The results showed that the experimental group improved its ability to decide, based on analytical thinking, about economic and healthcare-related everyday decision-making situations after they received the metacognitive strategy training. This improvement was maintained one month later. However, the ability to apply decision rules, which requires high cognitive effort, did not improve. In conclusion, it is important that some aspects of the analytical decision-making ability of amnestic MCI patients were improved due to the present metacognitive intervention.

## 1. Introduction

Mild cognitive impairment (MCI) is considered a stage between normal cognitive function and dementia ([1]; [48]). More precisely, the cognitive decline is greater than expected for the patient’s age and educational level. One or more cognitive functions are affected, including memory, attention, executive functions, language, and visuospatial skills. Older adults with a diagnosis of MCI, even those that revert to normal cognition, have a high risk of developing dementia ([52]). 

There are two main subtypes of MCI. Non-amnestic MCI is characterized by impairment in one or more non-memory domains, but preserved memory ([1]; [48]). The other subtype of MCI, which is the most common one ([47]), and which has been mostly considered a prodromal stage of Alzheimer’s disease ([48]), is amnestic MCI (aMCI). Single-domain aMCI patients present deficits only in memory, while multiple-domain aMCI patients present deficits not only in memory, but also in one or more non-memory cognitive functions ([48]). 

In MCI, criteria for a diagnosis of dementia are not met, because cognitive decline is not extensive, and daily functioning is not significantly affected ([1]; [48]). However, although MCI diagnosis requires that independence in daily functioning is preserved ([1]; [48]), difficulties are observed in more complex daily tasks, concerning financial issues ([26]) and health issues such as capacity to consent to medical treatment ([43]).

Decision-making is an essential ability of a person to live independently and successfully ([41]; [55]). Decision-making is the process of choosing between at least two competing options by analysing the costs and benefits of each option and estimating its future consequences ([40]). There are two classic behavioral task types to assess decision-making ability. In decision under ambiguity tasks explicit information about the outcomes and probabilities of their occurrence is not offered and should be learned through experience ([6]), while in decision under risk tasks this information is provided and should be used to evaluate options ([10]). 

Two different decision-making processes (or systems) described by dual process theories ([22]; [23]; [36]) are thought to be engaged during decision-making under risk and under ambiguity ([56]; [59]). Decision under ambiguity tasks require an experiential mode of thinking (or intuitive system or system 1), which is unconscious, automatic, effortless, and rapid, based on intuition and past experiences. On the contrary, decision under risk tasks require an analytical mode of thinking (or deliberative system or system 2), which is conscious, cognitively effortful, and slow, based on reasoning and analysis of information. 

MCI patients present deficits in both experiential and analytical processes. They show difficulties in the Iowa Gambling Task, which is a decision under ambiguity task ([4], [5]; [65]), as well as in the Probability-Associated Gambling Task ([65]), the Game of Dice Task ([59]; [66]), the Game of Dice Task-Double ([45]), which are laboratory computerized decision under risk tasks, and in a decision under risk task about real-life health-related situations ([46]). To our knowledge, there is only one decision-making intervention study targeting MCI patients. [13] ([13]) found that cognitive training on number processing and executive functions improved MCI patients’ performance on a decision-making under risk task as well as on a health-related ratio processing task.

### 1.1. Metacognition and Decision-Making Interventions

A theoretical model about decision-making that has practical implications for decision-making interventions is the Integrated Judgment and Decision-Making Model (IJDM; [16]). According to this model, there is a metacognitive system that monitors and controls the analytical processes, the experiential processes and the processes of a wisdom/expertise system which consists of schemas formulated by analytical and experiential processes. [16] ([16]) proposed that decision-making interventions should focus on analytically created schemas, which are steps to analyse problems (analytical mode of thinking) and to monitor and control the decision-making process (metacognition). If these schematic structures are repeatedly applied to make decisions, they are internalized as a part of the wisdom/expertise system and ultimately, they are activated automatically every time a decision situation emerges.

A metacognitive perspective in decision-making training was also adopted by [3] ([3]). University students improved their decision-making performance after receiving a “metacognitive strategy instruction”. This instruction was based on the [14] ([14]) metacognitive instruction model, which consists of four steps. Translation refers to understanding the problem, while integration focuses on gathering and organizing the necessary information ([3]; [14]). Then, solution planning and monitoring emphasizes finding an appropriate strategy to solve the problem and monitoring its application. Finally, solution execution is reaching a decision and then checking for errors or missed information.

The IJDM ([16]) and [3]’s ([3]) instruction were utilized by [53] ([53]) to create a “metacognitive-strategy decision-making training” for older adults. During training, participants practiced in answering a series of metacognitive-strategic questions to make decisions about hypothetical real-life situations. Additionally, they practiced in choosing the decision rule ([44]) that would be most suitable to apply in specific decision-making situations. This training enhanced older adults’ analytical mode of thinking in everyday decision-making contexts and their ability to apply decision rules.

### 1.2. The Present Study

Based on evidence of impaired analytical decision-making in MCI and on metacognitive approaches for improving decision-making, the aim of the present study was to examine whether the addition of a metacognitive strategy training can improve the decision-making abilities of older adult patients diagnosed with aMCI and attending cognitive and physical intervention programs in a day care center for Alzheimer’s Disease. 

The hypothesis was that aMCI patients’ decision-making ability in everyday contexts and their ability to apply decision rules will improve directly after the metacognitive training, in comparison to before such a training, and that this improvement will be maintained or increased one month after the metacognitive training is finished. In addition, aMCI patients’ decision-making ability in everyday contexts and their ability to apply decision rules will improve more after their participation in metacognitive training, in comparison to aMCI patients that attend only the classic intervention programs (cognitive and physical interventions) offered by the day care center.

## 2. Methods

### 2.1. Study Design and Procedure

The study followed an experimental design (Figure 1). The experimental group (EG), which was participating in cognitive and/or physical intervention programs as patients—visitors of a day care center for Alzheimer’s disease, received an additional metacognitive strategy decision-making training, which consisted of three sessions, one each week for three consecutive weeks. The active control group (CG) of aMCI patients was also attending the same cognitive and physical intervention programs offered by the day care center. There was a pre-testing and a post-testing session for both groups one week before and after the metacognitive intervention in the EG respectively. There also was a follow-up session only for the EG, one month after the post-testing session. During all testing sessions, the ability to make decisions in everyday decision-making situations and the ability to apply decision rules was assessed.

The first researcher conducted all the metacognitive strategy training sessions via a video conferencing program. Each testing session lasted 40 min to one and a half hours, depending on the participant, and each training session lasted one hour to one hour and a half. The EG took all sessions in groups of four to seven people, with a few exceptions that took some sessions alone or in groups of two, due to unexpected circumstances (e.g., illness). Most of the participants of the CG took both testing sessions one by one. 

During all testing sessions, the measures were administered using a Google form. Participants had to complete firstly the appropriate version of the Everyday Decision-Making Competence task ([53]; see below for a description), and then the appropriate version of the Applying Decision Rules task ([11]; see below for a description). There was no time limit for the completion of the form. The researcher provided explanations about the instructions of the measures, when she was asked, and help with technical difficulties. It should be noted that all measures and all decision-making problems that were used during training were previously pilot-tested.

### 2.2. Participants and Ethical Standards

The sample comprised 55 older adults with aMCI that were recruited from the Day Care Centre “Saint Helen” of the Alzheimer Hellas (DCCAH) via phone calls or while they were attending online physical intervention programs or cognitive training programs. Based on a power analysis that was conducted, using G*Power ([25]), a total sample size of 34 participants was recommended to detect an effect size of η^2^ = 0.25, with an alpha of 0.05 and to achieve a power of 0.80.

Inclusion criterion was a diagnosis of single-domain or multiple-domain aMCI. The diagnosis was given at most 10 months before the pre-test of the present study, by a consensus of specialized health professionals of the DCCAH, considered experts in neurocognitive disorders. A neurological examination, a neuropsychological and neuropsychiatric assessment, neuroimaging (computed tomography or magnetic resonance imaging), and blood tests were considered for diagnosis.

The criteria for the diagnosis of MCI were: (a) diagnosis of Minor Neurocognitive Disorders according to DSM-5 ([2]), (b) Mini-Mental State Examination ([27]; [29]) total score ≥ 24, (c) stage 3 of the disease according to Global Deterioration Scale ([51]), and (d) 1.5 standard deviation below the normal mean according to age and education, in at least one cognitive domain according to the utilized neuropsychological tests. In addition, the Montreal Cognitive Assessment Scale ([50]; [42]) was used to assess the general cognitive status, and the Functional Cognitive Assessment Scale ([38]) to assess the ability to organize and execute six different activities of daily living. Standardized tests for the assessment of general cognitive and functional abilities, memory capacity, language abilities, executive functions, and attention were used as well. The entirety of the neuropsychological tests included in the battery is presented in detail in [61] ([61]).

Exclusion criteria were a psychiatric illness or an untreated affective disorder (Major Depression/General Anxiety Disorder). Thus, the Geriatric Depression Scale ([28]; [64]) and the Beck Depression Inventory ([7]), the Short Anxiety Screening Test ([31]; [60]) and the Beck Anxiety Inventory ([8]) were used to exclude affective disorders and the Neuropsychiatric Inventory ([15]; [49]) to exclude neuropsychiatric symptoms. 

The EG consisted of 27 participants (18 women and 9 men) aged 63–79 years (*M* = 70.63, *SD* = 4.47). Twenty-three were given a diagnosis of multiple-domain aMCI and four a diagnosis of single-domain aMCI. Their years of education ranged from 6 to 20 (*M* = 13.44, *SD* = 3.95) and the years they have been attending programs at the DCCAH ranged from 1 to 12 (*M* = 4.11, *SD* = 3.69). One participant was receiving cholinesterase inhibitors, six were receiving antidepressants and five anxiolytics. 

The CG included 28 participants (21 women and 7 men) aged 62–80 years (*M* = 70.86, *SD* = 4.67). Twenty-five were given a diagnosis of multiple-domain aMCI and three a diagnosis of single-domain aMCI. Their years of education ranged from 6 to 21 (*M* = 13.71, *SD* = 3.71) and the years they have been attending programs at the DCCAH ranged from 1 to 13 (*M* = 4.50, *SD* = 3.47). Two were receiving cholinesterase inhibitors, six antidepressants and four were receiving anxiolytics. 

Participants were assigned randomly to the two groups. The two groups did not differ significantly in gender [χ^2^(1, 55) = 0.463, *p* = .496], subtype of aMCI [χ^2^(1, 55) = 0.208, *p* = .648], years of age [*F*(1, 53) = 0.034, *p* = .854], years of education [*F*(1, 53) = 0.068, *p* = .795] and years at the DCCAH [*F*(1, 53) = 0.162, *p* = .689].

Participants were informed about the aim and the procedure of the study both orally and via an informative email. In addition, since demographic data, which are considered personal data, were collected, the General Data Protection Regulation, which is the European Union law that exists since 25 May 2018 was applied. According to the law, the use of sensitive personal data is allowed only for research reasons. So, participants were also informed that their data would be kept confidential. Therefore, participants gave informed consent, agreeing that their participation was voluntary and that they could withdraw at any time, without providing a reason and without cost. The protocol of the study was approved by the Scientific and Ethics Committee of Alzheimer Hellas (Scientific Committee Approved Meeting Number: 82/19-10-2022) and followed the principles outlined in the Helsinki Declaration.

### 2.3. Measures

#### 2.3.1. Everyday Decision-Making Competence Τask

The Everyday Decision-Making Competence task (EDMC; [53]) was used to assess the decision-making ability in everyday situations. It consists of 12 decision-making problems about daily (four problems, e.g., decide from which supermarket to buy groceries), economic (four problems, e.g., decide which insurance policy to buy for a car) and healthcare (four problems, e.g., decide which therapy is best to treat hypothyroidism) scenarios (see Table 1). Half are analytical-based, which means that they require effortful analytical processing, such as doing mathematical calculations. The rest are experiential-based and present two conflicting options. One option is based on base-rate information provided by a reliable source of information, thus engaging analytical thinking. The other option is based on information about a single case or a personal experience and thus is chosen if experiential processing is preferred over reliance on base-rate evidence. The participants had to choose from a set of four possible answers [“Certainly (option A/B)”, “Probably (option A/B)”], so that scores ranged from 1 (indicating the disadvantageous or the experiential decision for the analytical-based and the experiential-based problems respectively) to 4 (indicating the advantageous or the analytical decision for the analytical-based and the experiential-based problems respectively). Total scores were computed for each of the two types of problems and for each of the three types of scenarios. The task has two versions that were translated in Greek. The follow-up version was created by changing some superficial information of the post-test version (i.e., names, objects and numbers).

#### 2.3.2. Applying Decision Rules Τask

The Applying Decision Rules task (ADR), which is a subtest of the Adult Decision-Making Competence battery ([11]), consists of 10 problems that evaluate the ability to correctly apply decision rules of varying complexity. For each problem, five electronic products with numeric ratings of their features (e.g., picture quality) are presented in a table. Participants must select one or more products by applying the decision rule that is described each time. Decision rules were elimination by aspects, satisficing, lexicographic, and equal weights ([44]; for a short description of each decision rule see the following section). The task’s final score was computed as the mean of correct answers. The task was translated in Greek and three versions were created, one for each time of assessment. The differences between them were superficial (i.e., computers, televisions and mobile phones on pre-test, post-test and follow-up assessment respectively, and names). 

### 2.4. Intervention 

During the first training session, the researcher stressed the difference between experiential and analytical thinking (e.g., [36]) during daily, economic, and healthcare decision-making. Four metacognitive-strategic questions (see Table 2) which are a simplified version of [3] ([3]) “metacognitive strategy instructions”, were introduced as an “analytically created schema” ([16]) to promote metacognitive and analytical thinking. These questions are the core of the present intervention as they were answered every time a decision problem was analyzed. During this session participants answered the questions to analyze two daily (one analytical-based and one experiential-based) and two healthcare (one analytical-based and one experiential-based) decision-making problems of the pre-test version of the EDMC ([53], see Table 1), as well as similar problems that participants were bringing up from their everyday life. Emphasis was given to the second metacognitive-strategic question, i.e., collecting and organizing all necessary information (see Table 2).

During the second training session, at first, a review of the previous session’s content was made, and participants practiced on answering the metacognitive-strategic questions to analyze two economic problems (one analytical-based and one experiential-based) of the pre-test version of the EDMC ([53]). During the rest of the session, the emphasis was given on the third metacognitive-strategic question. Specifically, four examples of everyday decision-making situations that provided instructions leading participants to apply a specific decision rule were presented (see Table 3). After each example was analyzed by applying the metacognitive-strategic questions, the researcher explained the relevant decision rule and then a discussion was made about some possible situations each rule could be applied in. 

The four decision rules used refer to heuristics defined as methods that reduce the cognitive effort associated with decision-making ([44]; [58]). Satisficing decision rule is choosing the first in order alternative that meets the predefined cutoff values for all features. Lexicographic decision rule is selecting the alternative with the highest value on the most important feature and then on the second most important feature if there is a tie and so on. Elimination-by-aspects decision rule refers to choosing the alternative that meets a cutoff value predefined for the most important feature and then for the second most important feature if there is a tie and so on. Finally, equal weights decision rule refers to choosing the alternative with the highest total value computed by summing the values of all features of the alternative (see Table 3).

At the beginning of the third session, at first, a review about the previous sessions’ content was made. Then, participants answered the metacognitive-strategic questions for four examples of everyday decision-making situations which provided instructions that were leading them to apply a specific decision rule. The examples were more complex than the ones analyzed in the second session, because they had more than one characteristic of the possible choices to take into account. Finally, two analytical-based (one daily and one healthcare) and two empirical-based (one economic and one healthcare) decision-making problems of the pre-test version of the EDMC ([53]) were analyzed using the metacognitive-strategic questions.

### 2.5. Statistical Analyses

The data analysis was conducted in SPSS version 29. After computation of the EDMC mean total scores (experiential-based, analytical-based, daily, economic and healthcare scores) and the ADR scores (see Table 4), repeated measures ANOVAs were conducted to compare the performance of the EG in the three times of assessment. Subsequently, mixed-design 2 × 2 ANOVAs (representing pre-test and post-test measurements × two groups, EG and CG) were conducted to examine the main and the interaction effects of group and time of assessment on performance. In a third step, given that the CG had higher performance in the pre-tests, compared to the EG, we proceeded to ANCOVAs, using as independent variable the group (EG, CG), as dependent variable the performance in each task in the post-test measurement, and the pre-test measurement as the covariate variable. Only the statistically significant results of the ANCOVAs will be mentioned. Partial eta-squared (η_p_^2^) was used for the estimation of the effect size. Finally, to control for multiple testing, a Bonferroni correction was applied, i.e., significant *p* = .5/6 = .008.

## 3. Results

### 3.1. Experimental Group: The Effects of Time of Assessment (Pre-, Post-Test, and Follow Up) on the Performance

A significant effect of the time of assessment was found on both EDMC analytical-based scores, *F*(2, 52) = 10.254, *p* < .001, η_p_^2^ = 0.28, and EDMC experiential-based scores, *F*(2, 52) = 14.299, *p* < .001, η_p_^2^ = 0.36. Specifically, both analytical-based scores, I-J = −0.364, *p* = .005, and experiential-based scores, I-J = −0.648, *p* < .001, increased between the pre-test assessment and the post-test assessment. In addition, a significant increase of both analytical-based scores, I-J = −0.414, *p* < .001, and experiential-based scores, I-J = −0.716, *p* < .001, was found between the pre-test assessment and the follow-up assessment. However, there was no significant increase of either score types between the post-test assessment and the follow-up assessment.

A significant effect of the time of assessment was also found on the EDMC daily scores, *F*(2, 52) = 20.321, *p* < .001, η_p_^2^ = 0.44 and economic scores, *F*(2, 52) = 9.830, *p* < .001, η_p_^2^ = 0.27, as well as a same trend as regards the time of assessment and healthcare scores, *F*(2, 52) = 4.098, *p* = .022, η_p_^2^ = 0.14. Specifically, between the pre-test assessment and the post-test assessment, there was a significant increase of daily scores, I-J = −0.639, *p* < .001, and economic scores, I-J = −0.537, *p* < .001, but not of healthcare scores. Furthermore, between the pre-test assessment and the follow-up assessment, a significant increase of daily scores, I-J = −0.750, *p* < .001, economic scores, I-J = −0.519, *p* = .008, as well as a trend of increase of healthcare scores, I-J = −0.426, *p* < .033, was found. However, there was no significant increase of none of the score types between the post-test assessment and the follow-up assessment. The effect of time of assessment on ADR scores was not significant (see Table 4, Figure 2).

### 3.2. Comparison of the Experimental Group and the Active Control Group: The Effects of Group and Time of Assessment (Pre- and Post-Test) on Performance

Mixed-design ANOVA revealed a significant interaction effect (time of assessment × group) on EDMC analytical-based scores, *F*(1, 53) = 8.702, *p* = .005, η_p_^2^ = 0.141, suggesting that the pattern of change in analytical-based scores from the pre-test to the post-test assessment differed between the two groups. Specifically, as depicted in Figure 3, the EG, although it performed lower than the CG at the pre-test assessment, it showed improved post-test performance and outperformed the CG, while the CG showed slightly decreased post-test performance. The main effects of the time of assessment, *F*(1, 53) = 2.779, *p* = .101, and the group, *F*(1, 53) = 0.616, *p* = .436, were not significant.

A similar trend (time of assessment × group), *F*(1, 53) = 4.445, *p* = .04, η_p_^2^ = 0.077, and a significant main effect of the time of assessment, *F*(1, 53) = 25.787, *p* < .001, η_p_^2^ = 0.327, were found on EDMC experiential-based scores, indicating that the pattern of change in the experiential-based scores from the pre-test to the post-test assessment tends to vary between the two groups. Specifically, as depicted in Figure 4, the EG, although it performed lower than the CG at the pre-test assessment, showed improved post-test performance and outperformed the CG. The main effect of the group was not significant, *F*(1, 53) = 0.000, *p* = .995. Regarding the main effect of the time of assessment, Bonferroni pairwise comparisons showed improved post-test EDMC experiential-based scores for both groups (EG and CG), I-J = −0.458, *p* < .001. The subsequent ANCOVA revealed that there was a trend of the covariate (pre-test EDMC experiential-based score) to affect the post-test EDMC experiential-based score, *F*(1, 52) = 5.012, *p* = .029, η_p_^2^ = 0.088. When this trend was taken into account, only a trend of difference between the post-test EDMC experiential-based score of the two groups remained, *F*(2, 52) = 3.399, *p* = .041, η_p_^2^ = 0.116.

A significant main effect of the time of assessment was found on EDMC daily scores, *F*(1, 53) = 32.132, *p* < .001, η_p_^2^ = 0.377, indicating an improvement from the pre-test to the post-test assessment for both groups. The main effect of the group, *F*(1, 53) = 1.42, *p* = .239, and the interaction effect (time of assessment × group), *F*(1, 53) = 3.014, *p* = .088, were not significant. Regarding the main effect of the time of assessment, Bonferroni pairwise comparisons showed improved post-test EDMC daily scores for both groups (EG and CG), I-J = −0.489, *p* < .001.

A significant interaction effect (time of assessment × group), *F*(1, 53) = 14.178, *p* < .001, η_p_^2^ = 0.211, and a trend of the time of assessment to affect performance, *F*(1, 53) = 4.435, *p* = .04, η_p_^2^ = 0.077, were found on EDMC economic scores, indicating that the pattern of change in experiential-based scores from the pre-test to the post-test assessment varied between the two groups. Specifically, as depicted in Figure 5, the EG, although it performed lower than the CG at the pre-test assessment, showed improved post-test performance and outperformed the CG. The main effect of the group was not significant, *F*(1, 53) = 2.189, *p* = .145.

The main effect of the time of assessment, *F*(1, 53) = 2.865, *p* = .096, the main effect of the group, *F*(1, 53) = 0.795, *p* = .376, and the interaction effect (time of assessment × group), *F*(1, 53) = 1.37, *p* = .247, on the EDMC healthcare scores were not significant.

The main effect of the time of assessment, *F*(1, 53) = 3.838, *p* = .055, the main effect of the group, *F*(1, 53) = 1.622, *p* = .208, and the interaction effect (time of assessment × group), *F*(1, 53) = 0.017, *p* = .898, on the ADR scores were also not significant.

## 4. Discussion

The purpose of the present study was to assess the effectiveness of a metacognitive strategy training on aMCI patients’ decision-making abilities. According to the hypothesis of the study, aMCI patients’ ability to make decisions about everyday situations and their ability to apply decision rules would improve directly after the metacognitive intervention and this improvement would be maintained or increased one month later. Furthermore, aMCI patients’ decision-making ability in everyday contexts and their ability to apply decision rules would improve after training in comparison to an active control group of aMCI patients. The hypothesis was partially confirmed. 

### 4.1. Decision-Making Ability in Everyday Situations

Results showed that aMCI patients’ performance on analytical-based and experiential-based problems of the EDMC, was increased after the intervention, in comparison to before the intervention. Importantly, these benefits were maintained one month after the post-test assessment. In addition, the effectiveness of the metacognitive intervention was also found when the two groups were compared. The findings indicate that aMCI patients learned to rely more on analytical thinking while making decisions not only about analytical-based problems that require the analysis of the problem’s structure and thus increased cognitive effort, but also about experiential-based problems that require rejection of experience-based information and reliance on base-rate information ([53]). In line with the IJDM ([16]), the present metacognitive training enhanced the use of the analytical system during decision-making, because of the repeated application of an “analytically created schema”. This schema was internalized as a part of the wisdom/expertise system and participants could use it consciously or unconsciously to make advantageous decisions. In addition, these results are in agreement with [3] ([3]), who found an improvement in decision-making ability of university students after a “metacognitive strategy instruction”. Finally, like in the present study, [53] ([53]) combined IJDM ([16]) and [3] ([3]) instruction and created a “metacognitive-strategy decision-making training” that improved decision-making abilities of older adults.

Analytical decision-making requires executive functions and working memory capacity to categorize information, evaluate options and monitor the application of strategies (e.g., [10]; [57]; [56]). However, aMCI patients present deficits in executive functions and working memory ([37]). Therefore, it is unsurprising that MCI patients make more disadvantageous decisions than healthy peers in risk situations, which require analytical thinking ([56]) and which have been associated with executive functions and ratio processing abilities ([45]). Even though cognitive deficits affect analytical decision-making, in the present study aMCI patients’ analytical decision-making ability has improved. Thus, it can be assumed that the metacognitive system can compensate for the effect of cognitive deficits in analytical thinking. Specifically, during the present intervention, participants’ metacognitive system was improved by obtaining declarative knowledge about decision-making process (the analytically created schema and the decision rules). Additionally, their ability to regulate their thinking process (metacognitive control) while making decisions was enhanced by using the analytically created schema.

In regard to the performance on the economic decision-making scenarios, the results indicated that it improved after the metacognitive intervention, in comparison to before the intervention and in comparison to the CG of aMCI patients. This could be explained by the fact that more economic decision-making problems were discussed during the intervention. The majority of the problems participants were bringing up from their everyday life, as well as the decision-making problems that provided instructions to apply a decision rule, were mostly economic. 

Additionally, as far as the performance on the EDMC healthcare scenarios is concerned, it was not improved after the metacognitive intervention in comparison to before the intervention and in comparison to the CG. Older adults with an aMCI diagnosis face a plethora of health-related decision-making situations in their everyday life. Therefore, according to IJDM ([16]), it is possible that participants had already formed schemas about decision-making in healthcare situations, by accumulating similar decision-making experiences. As this kind of schemas is “more resistant to change” ([16]), training effects were not observed immediately after the intervention. However, an increased performance on healthcare scenarios was found on the follow-up assessment in comparison to the pre-test assessment of the EG, which means that participants were making more advantageous decisions about health-related issues a month after the post-test. This probably indicates that participants abandoned their previous schemas, only after further repeated application of the “analytically created schema”, that they were taught during the intervention, in their daily life after the completion of the intervention.

Finally, aMCI patient’s decision-making ability about daily scenarios was improved after the intervention in comparison to before the intervention, but not in comparison to the CG. Thus, it cannot be stated with certainty that it was the metacognitive intervention that caused this improvement. Probably participants had already formed schemas about decision-making in daily situations which were effective and thus they were maintained even after the intervention.

It should be noted that participants of the present study were attending cognitive training and/or physical intervention programs at the same time period that they were participating in the present study, as visitors of DCCAH. In the future, metacognitive decision-making interventions should be applied in MCI patients that do not attend other programs- if ethically possible, so that any improvement in decision-making abilities due to this specific type of intervention will be measured in its real magnitude.

### 4.2. Ability to Apply Decision Rules 

Concerning performance on the ADR ([11]), results did not show a significant improvement after the intervention and in comparison to the CG. Like the EDMC ([53]), the ADR ([11]) is a task that requires analytical thinking, since comparisons of the values, information integration, and suppression of irrelevant or no longer relevant stimuli are needed ([18]). However, in comparison to the EDMC, the ADR is more cognitively effortful, as it consists of five possible alternatives and an arithmetic value for each one of five features of each alternative. Moreover, some items of the task require the application of two or three decision rules in a predefined order. So, the ADR places high demands on fluid cognitive abilities ([11], [12]), working memory ([19], [20], [17]; [54]) and the inhibition dimension of executive functioning ([18], [19]). Therefore, aMCI patients did not improve in this task, probably because the practice on applying decision rules was not adequate and not with material as complex as the ADR, or because their cognitive deficits played more vital role than metacognition, while applying decision rules. However, as participants of the present study were already receiving cognitive training in the DCCAH, the effectiveness of an intervention targeting more intensely the cognitive functions that seem to influence the ability to apply decision rules should probably be more extensively studied in the future. 

### 4.3. Neuropsychology of Decision-Making

According to a review by [30] ([30]), there is a complex neural network that supports decision-making. The main brain areas of this network are the orbitofrontal cortex and the ventromedial prefrontal cortex (stimulus encoding system), the anterior cingulate cortex and the lateral prefrontal and parietal cortices (action selection system), the basal ganglia, the amygdala and the insula (expected reward system). [62] ([62]) have also highlighted the significant role of frontal lobes in the identification of the decision options, and the construction of the options’ values (ventromedial and dorsomedial frontal lobe). In addition, anodal transcranial direct current stimulation to the right dorsolateral prefrontal cortex (DLPFC) has been associated with an increase in analytical judgment and decision-making ([21]), and with creative problem-solving, by promoting convergent analytical thinking ([34]). Finally, analytical decision-making under risk performance has been associated with dorsolateral and ventromedial prefrontal cortices’ activity ([30]).

Most of the above mentioned brain areas seem to be negatively affected in MCI. In particular, MCI patients present changes in the orbitofrontal cortex (e.g., [24]), the anterior cingulated cortex (e.g., [9]), the DLPFC (e.g., [39]), the parietal lobe (e.g., [35]), the basal ganglia, the amygdala (e.g., [63]) and the insula (e.g., [24]). However, some of the brain areas involved in decision-making are probably less impaired than others in aMCI. So, maybe the decision-making ability in everyday situations was improved, because it is based on the less impaired brain areas’ activity, which was probably enhanced during the present intervention. Also, it can be assumed that the ability to apply decision rules was not improved, because the brain areas that are involved in this decision-making ability are significantly impaired in aMCI. It should be noted that financial capacity in aMCI has been predicted by the volume of the angular gyrus ([32]) and that medical decision-making capacity in aMCI patients has been associated with posterior cortical brain metabolism of N-acetylaspartate/Creatine ([33]).

### 4.4. Limitations and Future Research

This study has limitations that future research should address. Firstly, the EDMC is not as complex as real-life decision-making situations can be, because all necessary information is given in the problem description, there are only two options every time and no time limit. Thus, future research could use additional tasks that resemble real-life decision-making situations even more. In addition, the short duration of the intervention (only 3 sessions) and the relatively small time period between the post-test assessment and the follow-up assessment should be taken into account in future studies. Finally, longitudinal designs, neurophysiological measures and other types of MCI should be considered in future research.

## 5. Conclusions

In conclusion, as MCI is associated with deficits in analytical decision-making (e.g., [59]), an ability highly relevant to daily functioning ([41]), the present study’s aim was to examine the efficacy of a metacognitive strategy training on aMCI patients analytical decision-making abilities. Significant improvement was observed in the ability to think analytically while making decisions about everyday situations, economic and healthcare-related, but not in the ability to apply decision rules, which is a cognitively demanding analytical ability. The present study is an important step in examining decision-making interventions for aMCI.

## Figures and Tables

**Figure 1 jintelligence-11-00182-f001:**
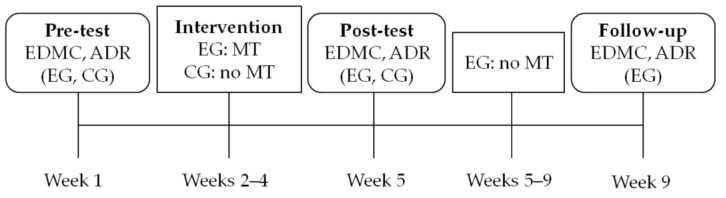
Study design. Note: EDMC = Everyday Decision-Making Competence task; ADR = Applying Decision Rules task; EG = Experimental Group; CG = Active Control Group; MT = Metacognitive Strategy Training; It must be noted that all aMCI patients of the EG and the CG were attending the classic interventions offered by the day care center (cognitive and physical training) during all these weeks.

**Figure 2 jintelligence-11-00182-f002:**
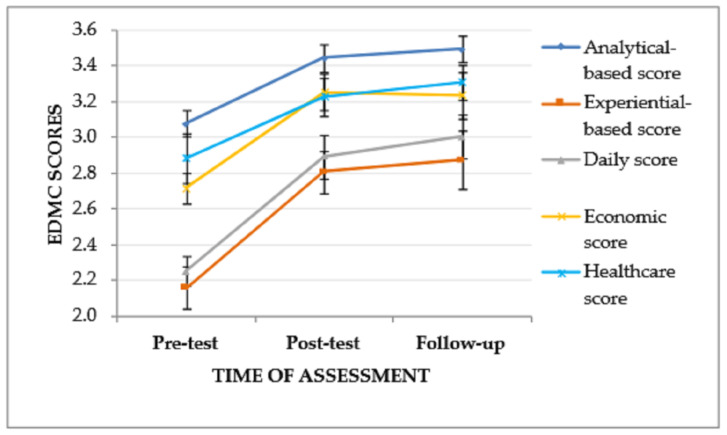
EDMC scores of the experimental group in the three times of assessment. Error bars represent standard errors of the means.

**Figure 3 jintelligence-11-00182-f003:**
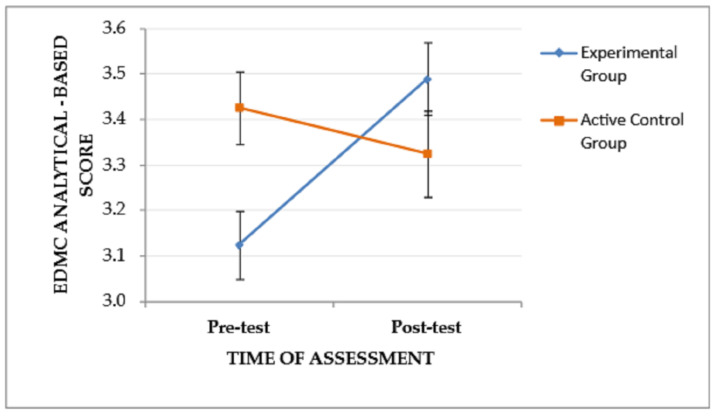
The effect of the group and the time of assessment on the EDMC analytical-based score. Error bars represent standard errors of the means.

**Figure 4 jintelligence-11-00182-f004:**
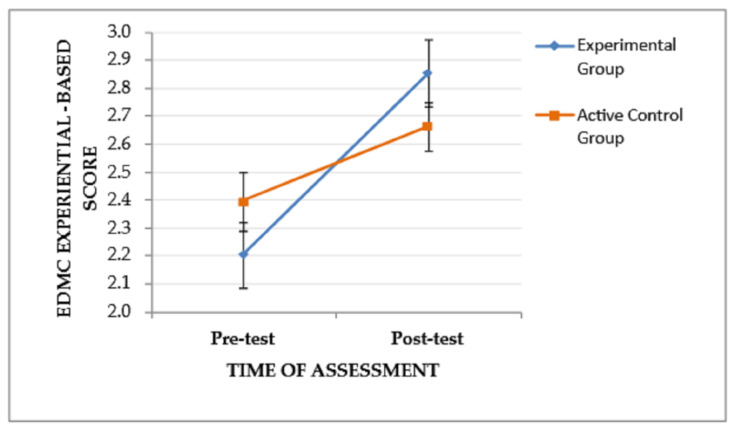
The effect of the group and the time of assessment on the EDMC experiential-based score. Error bars represent standard errors of the means.

**Figure 5 jintelligence-11-00182-f005:**
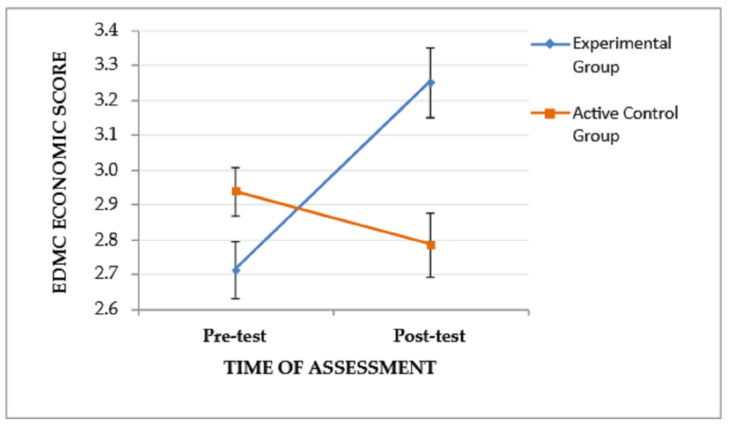
The effect of the group and the time of assessment on the EDMC economic score. Error bars represent standard errors of the means.

**Table 1 jintelligence-11-00182-t001:** Pre-Test Everyday Decision-Making Competence Τask Items Examples.

Type of Problem	Example
Analytical-based	Dimitra lives in the suburbs. She started a new job in the city center and must use the city bus from Monday to Friday at 7:30 a.m. and 6:10 p.m. The one-way ticket costs 1 euro. Dimitra found out that there are two kinds of bus passes and has to choose between buying the monthly bus pass which costs 25 euro, or the three-month bus pass which costs 60 euro. Which card is Dimitra most likely to buy? The monthly pass—1 point Perhaps the monthly pass—2 points Perhaps the three-month pass—3 points The three-month pass—4 points
Experiential-based	Konstantinos wants to give his friend a pack of tennis balls as a Christmas present. Tennis experts writing in the magazine “All about tennis” suggest the “STAR” balls. As soon as Konstantinos arrives at the sports shop, he overhears two tennis players saying that “MEGA” balls are better than “STAR” balls. The “STAR” balls are in a pack of six at a cost of 18 euro. At the same time, “MEGA” balls are in a pack of six at a cost of 18 euro, or in a pack of twelve at a cost of 36 euro. Which brand of tennis balls is Konstantinos most likely to buy for his friend? The “MEGA” balls—1 point Maybe the “MEGA” balls—2 points Maybe the “STAR” balls—3 points The “STAR balls”—4 points

**Table 2 jintelligence-11-00182-t002:** Metacognitive-Strategic Questions.

Metacognitive-Strategic Questions
1. Do I understand what the problem/question is? (If not, I reread/re-think about it until I understand it.)
2. What is the basic information of the problem that I need to decide? I describe it in my own words. (Do I have all the information necessary to decide? If not, what additional information do I need to decide?) How can I organize this information?
3. What is the strategy/the decision rule I need to follow to decide? Is this the most appropriate one? If not, I implement some other.
4. What is my decision? I check to see if I missed some information or if I made a mistake. Have I made the right decision?

**Table 3 jintelligence-11-00182-t003:** Examples of Decision Rules Problems.

Decision Rule	Example
Satisficing (session 2)	You want to learn Spanish and you decide to search online for Spanish language schools in your area. But you are in a hurry, so you decide to choose the first one you see that charges up to 15 € per hour. You find the following four language schools: “Callisto”, which has the best reviews and charges 17 € per hour. “Madrid”, which is the most popular and charges 16 € per hour. “Goal” which is located 40 m away from your home and charges 15 € per hour. “Gnostis” which gives emphasis on oral practice and charges 14 € per hour. Which school will you choose?
Lexicographic (session 2)	You want to go on holiday with your friend in Corfu and you want to find the hotel with the best reviews. You search on various websites and find the following four rooms available for the period you are interested in: The room at hotel “Ariston” with sea view and a rating of 6/10 The room at the hotel “Karavi” with free breakfast and a rating of 4/5 The room at the hotel “Phoenix” which is close to the center of Corfu and has a rating of 3/5 The room in hotel “Helios” with swimming pool and rating 7/10 Which room will you choose?
Elimination-by-aspects (session 3)	Serious damage has occurred to the plumbing in your home. Your neighbors recommend four plumbers, and you decide to talk to all four before making a decision. For you, the most important thing is that you don’t have to pay more than 500 €. If the plumbers who will charge you 500 € or less are more than one, then you choose the one who will finish the job in 10 days or less. The plumbers are the following: John, who asks for 520 € and will take one week. Nicolas, who asks for 400 € and will take two weeks. Gregory, who asks for 500 € and will take one week. Dimitris, who asks for 520 € and will take 6 days. Which plumber will you choose?
Equal weights (session 3)	You want to take your friends out to dinner for your birthday and you want to book a table at the restaurant with the highest average score for food, service, and music. The taverns you find on a website are: “Promenade” with a rating of 4/5 for food, 2/5 for service and 3/5 for music “Ocean” with a rating of 3/5 for food, 4/5 for service and 4/5 for music “Guitar” with a rating of 2/5 for food, 2/5 for service and 4/5 for music “Starfish” with a rating of 4/5 for food, 3/5 for service and 3/5 for music Which tavern will you choose?

**Table 4 jintelligence-11-00182-t004:** Mean and Standard Deviation of the EDMC and ADR Scores as a Function of Group and Time.

Type of Score	Experimental Group						Active Control Group			
	Pre-Test		Post-Test		Follow-Up		Pre-Test		Post-Test	
	M	SD	M	SD	M	SD	M	SD	M	SD
EDMC										
Experiential-based	2.15	0.61	2.80	0.62	2.87	0.84	2.35	0.55	2.61	0.46
Analytical-based	3.07	0.39	3.44	0.41	3.49	0.38	3.38	0.42	3.27	0.51
Daily	2.25	0.44	2.89	0.64	3.00	0.64	2.53	0.52	2.87	0.41
Economic	2.71	0.43	3.25	0.51	3.23	0.69	2.94	0.36	2.79	0.49
Healthcare	2.88	0.73	3.22	0.56	3.31	0.52	3.12	0.55	3.18	0.54
ADR	0.31	0.17	0.36	0.21	0.37	0.24	0.26	0.17	0.30	0.20

Note: EDMC = Everyday Decision-Making Competence task; ADR = Applying Decision Rules task.

## Data Availability

Data available upon duly justified request.

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
