# Peer review of "Metacognitive Strategy Training Improves Decision-Making Abilities in Amnestic Mild Cognitive Impairment"

_jintelligence, 2023, doi:10.3390/jintelligence11090182_

Round 1
Reviewer 1 Report
The manuscript is related to a rather actual topic - decision-making in mild cognitive impairment on the background of learning a metocognitive strategy. The authors took as an experimental group patients with Alzheimer's disease. However, this can only be guessed at. While some sections of the paper are very detailed, others need to be improved:
1. The abstract should be improve. Although there are no requirements for abstract structuring. But most readers look only this information. Therefore, the methods used, a brief description of the subjects, the most significant results and a significant conclusion should be reflected here.
2. INTRODUCTION. Since this topic is related not only to Alzheimer's disease, but also to a line of neurodegenerative diseases (Parkinson's disease, Lewy bodies, Multiple Sclerosis, etc.), the introduction will increase the significance of the publication if the authors show the significance of their development for the entire group of these diseases and show the goal - the disease Alzheimer's.
3. METHODS. As I said above about patients, one can only guess. Since there are few patients selected, a more detailed description should be given (anamnesis, leading cognitive impairments, etc.). Perhaps in the form of a separate table. What is an active control group?
4. RESULTS. Against the background of a detailed description of the methods, the results are presented insufficiently, only in the form of tabl. 4. This chapter should be expanded and structured by methods. It is possible to add pictures.
5. DISCUSSION. This chapter looks good. To improve the assessment of the work, it is possible to add to the discussion works with the factor structure of cognitive impairment in neurodegenerative diseases and the development of a strategy on this basis.
6. CONCLUSION It should be modified and structured according to the results for each method.
Reviewer 2 Report
I read with interest the manuscript entitled “Metacognitive Strategy Training Improves Decision-Making Abilities in Amnestic Mild Cognitive Impairment”. The manuscript is well-written and the hypothesis intriguing. However, there are two main methodological aspects that need to be changed or detailed.
The first is regarding the analysis. One could affirm that a specific treatment has improved over a control group only if the results of the interaction of the 2x2 ANOVA were significant (in the right direction obviously). For each dependent variable, this analysis must be performed. Second, why did the authors use EDMC pretest problems for training? As a result, the results could be contaminated since the test measures the precise context of the problems used during training.
Minor
Recommendations are not useful in the abstract. Please modify. If the results change, the abstract must also be updated.
The MCI description is too short, especially if compared to those of decision making. Please expand.
On the second page, please make a single hypothesis since the two mentioned were overlapped and part of the same method.
Please reconstruct all the section structure:
2.1 participants + ethical standard
2.2 Measures
2.3 Interventions
2.4 Procedure and study design
2.5 Statistical analyses
Figure 1 should be redrawn with a better timeline and each intervention at a specific point in time.
In the participant section, summarize the ethical standards. Please report GDPR privacy.
Please insert a Power analysis in order to determine the minimum sample size required.
Line 113 please change videotelefony with video meeting or a more appropriate term.
Why does only the experimental group attend the follow-up session?
Please explain why the experimental session took 1h to 1h30 (variability)?
Why did the control group take individual sessions but the experimental group took group sessions?
Line 152, do the authors mean MMSE <24.
How were MMSE and MoCA used/interchanged and their scores used?
I would appreciate it if you explained the criteria used for inclusion, which are used for experimental measurement, in greater detail.
For training, why did the author use the problems from the pretest of EDMC? In light of this, the possibility of absence of generalization became relevant.
Please divide the citation from other rules descriptions in parenthesis or describe the rules more clearly in lines 214-216.
Since the authors have 3 sessions in the experimental group and 2 in the control group, the design is unbalanced. A 2x2 mixed ANOVA (sessions with 2 levels and a group with 2 levels) without follow-up for the experimental group is recommended for each test. Look only at the interaction between factors. After this fundamental analysis, other analyses for including the follow-up are possible, like t-tests or one-way ANOVAs (but post-hoc are required). The use of MANOVA is not recommended because it does not permit to find specific areas in which the training is effective and areas in which it is not effective.
Discussion should consider all the above changes.
Reviewer 3 Report
The manuscript "Metacognitive Strategy Training Improves Decision-Making Abilities in Amnestic Mild Cognitive Impairment" is well written and presents a study on a group of aMCI patients (n=55).
The data is from a large scale study, investigating aMCI more broadly. In this manuscript the authors report on two tasks assessing decision-making. The design cannot address whether metacognitive therapy improves decision-making (see claim on line 319ff) as one expects a learning effect (repeated exposure to same task, just replacing a few words won't remove the general familiarity with the test). To address it, a non-aMCI group and their improvement from pre to post test would be needed. The percentage increase in a non-aMCI group can then be compared to the aMCI groups (with / without Metacogn training intervention), i.e., is there more or at least equal improvement in the experimental group as in the non aMCI group? To some extent the active control group can take on this role too, i.e., is the percentage improvement larger in the experimental than in the control group? So, the design can well work. Here, the authors need to do a between(group) by within (pre-post) analysis and report the interaction between group and time point. Is the slope steeper (more improvement) in the experimental group than the control group? From table 4 it is 130% increase for the exp group but only 110 % for the control group (experiental based EDMC)
this 2x2 (pre vs post x experimental vs control group) analysis has to be done for each of the 6 tasks, and then controlled for multiple testing (false discovery rate), or pooling of daily, economic and healthcare as no rational is provide in the introduction why you expect a difference. That would then be only 3 tests (experiential, analytical, ADR).
thus, before the paper can be published, the authors have to report the interaction effect (if it is significant, the main effects are not of interest, see https://stats.stackexchange.com/questions/91074/the-main-effect-will-be-non-significant-if-the-interaction-is-significant)
A strength is that the authors assessed performance one month later, but unfortunately only in one group, and they continued with CT and PT (is that not true for the other group too?). Still, given that performance can decay or the intervention (strategy learned) be forgotten, to see no decline is positive. But it may also indicate a ceiling effect, hard to know given the chosen design.
I am aware that there are design limitations, but the proper statistical test should be used and based on those results, claims about supporting or not of ones hypothesis made.
minor
line 365 space missing between "observed immediately"
Round 2
Reviewer 1 Report
I saw in the text all the corrections to my comments. The manuscript may be accepted in its present form.
Author Response
Point 1: I saw in the text all the corrections to my comments. The manuscript may be accepted in its present form.
Response 1: Thank you again for all the comments.
Reviewer 2 Report
Dear Authors,
I reviewed the second version of the manuscript. Substiantially it is improved and all suggestions were made. Since new graphs were added, for a perfect scientific soundness, it is better to add the Standard error of the mean (SEM) bars in the graphs and remove the black border outside. It is a simple task that improve drammatically the quality of the figures. SEM values could be obtined in SPSS directly and (personalised) bars added in excel.
Author Response
Point 1: Since new graphs were added, for a perfect scientific soundness, it is better to add the Standard error of the mean (SEM) bars in the graphs and remove the black border outside. It is a simple task that improve drammatically the quality of the figures. SEM values could be obtined in SPSS directly and (personalised) bars added in excel.
Response 1: Thank you very much for your comment. We added SD, but we didn’t remove the black border because we observed that in other articles of this journal the border is kept as it is.
Reviewer 3 Report
Thank you for replying to my concerns and questions.
Was any in the CG receiving antidepressants or anxiolytics? If not, then this may explain the group difference at pre-test, i.e., those who receive antidepressants and anxiolytics might perform worse than those not receiving any drugs - did you check for that? Did you use drug use as co-variate (regress it out)?
some minor issues
line 33/34 "... it is cognitive decline greater ... " should be "... the cognitive decline is greater ..."
line 36: "MCI older adults" should be Older adults with MCI
line 39: domain (not domains)
line 42: is amnestic MCI (aMCI) Single <- . missing after (aMCI)
line 44: function (not functions)
line 312: Bonferroni correction (Bonferroni correction - Wikipedia) is by number of tests you do (5 from EDMC, 1 from ADR). It is common to refer to it as "a Bonferroni correction was applied, i.e., significant p = 0.5/6 = .008" - note: your formulation is not wrong but unusual.
figure 2-5: y-axis numbers need to be as 2.5 not 2,5 and so on
please also include SEM or SD not just mean in the graphs
Round 3
Reviewer 2 Report
I suggested to put the SEM bars in the graph as usually done in scientific papers. You have added SD bars, that appears misleading in their size. If it possible change the size of the bars. SEM = SD / SQRT(n).
Author Response
Point 1: I suggested to put the SEM bars in the graph as usually done in scientific papers. You have added SD bars, that appears misleading in their size. If it possible change the size of the bars. SEM = SD / SQRT(n).
Response 1: Thank you very much for your comment. We added SEM bars as you suggested.
Reviewer 3 Report
Thanks for providing the figures with SDs (SEM would look better).
Author Response
Point 1: SEM would look better.
Response 1: Thank you for your comment. We added SEM bars.